# Variation in the Diet of Hatchling Morelet’s Crocodile (*Crocodylus moreletii*) in the Wild

**DOI:** 10.3390/ani14172610

**Published:** 2024-09-08

**Authors:** Mariana González-Solórzano, Marco A. López-Luna, Laura T. Hernández-Salazar, Edgar Ahmed Bello-Sánchez, Jorge E. Morales-Mávil

**Affiliations:** 1Biología de la Conducta, Instituto de Neuroetología, Universidad Veracruzana, Dr. Luis Castelazo s/n Col. Industrial Ánimas, Xalapa 91190, Veracruz, Mexico; mgs_1503@hotmail.com (M.G.-S.); terehernandez@uv.mx (L.T.H.-S.); ebello@uv.mx (E.A.B.-S.); 2División Académica de Ciencias Biológicas, Universidad Juárez Autónoma de Tabasco, Carr. Villahermosa-Cárdenas km 0.5 s/n Entrq. Bosques de Saloya, Villahermosa 86039, Tabasco, Mexico; marco.lopez@ujat.mx

**Keywords:** crocodilia, dietary pattern, ontogenetic change

## Abstract

**Simple Summary:**

Crocodiles are one of nature’s most successful predators. Their prey varies according to their size. Adults are known to consume different types of vertebrates; juveniles have a similar diet, although their prey is smaller. However, hatchlings have a diet based mainly on insects. This study aimed to know the diet of hatchling Morelet’s crocodiles (*Crocodylus moreletii*) at a growth stage corresponding to their transition to the juvenile stage. Therefore, it is important to know which vertebrate prey are beginning to appear during this developmental change in crocodiles. To find out, we captured and washed the stomachs of 31 hatchling crocodiles in the Laguna de las Ilusiones in Tabasco, Mexico. Our results showed that crocodiles have a generalist diet, with prey ranging from invertebrates (mainly coleoptera and hemipterans) to some vertebrates (mostly fish and birds). It is important to know these changes in the diet of crocodilians as they develop to help us better interpret their natural history and propose better strategies for their study.

**Abstract:**

The relationship between diet and behavior is essential to understanding an animal’s strategies to obtain food, considering ontogenical changes. In reptiles, there is a relationship between the length of the individual and the size of the prey it consumes. Studies have focused on the ontogenetic changes in reptile diets from hatchling to adult, but only a few studies have focused on the transition from hatchling to juvenile. We aimed to describe and analyze the composition, variation, diversity, and overlap in the diet of hatchling Morelet’s crocodiles (*Crocodylus moreletii*) for three size intervals during the hatchling–juvenile transition. We captured 31 hatchling Morelet’s crocodiles in an urbanized lagoon in Tabasco. We performed stomach-flushing to determine the diet. Additionally, we estimated the volume, frequency of occurrence, and relative importance of diet items and analyzed the relationship between prey type and the total length of the individuals. The diversity of the hatchling prey suggests a generalist diet. We observed two items not previously described in the diet of hatchling crocodiles. In addition, we found differences in diet between the initial and final size intervals, as increases in the length of prey appeared that they did not consume when they were hatchlings. Our results contribute new information to the dietary changes that occur during the hatchling–juvenile transition.

## 1. Introduction

Studies on feeding habits in reptiles have helped to understand fundamental ecological aspects of the life history of organisms, establishing a relationship between diet and the strategies that the animal develops to obtain food, considering changes in its body size, physiology, and requirements [1].

In crocodiles, in general, offspring feed on arthropods (insects, arachnids, and crustaceans, among others) and fish. Juvenile crocodiles consume fish and small vertebrates, while adults include larger fish and vertebrates in their diet [2,3]. The heterogeneity of the diet in crocodiles is related to the size of the prey. Individual length in reptiles is positively related to their prey’s size [4,5,6,7].

Although various studies must try to determine the components of the crocodile diet and the ontogenetic changes that occur from hatchlings to adults [2,3,8,9,10]. Few studies have focused on specifying the diet of hatchlings during their transition to juveniles [2,11].

Therefore, this present study aims to describe the composition of wild Morelet’s crocodile (*Crocodylus moreletii*) hatchlings’ diet and the variation in diet during the hatchling–juvenile transition. We asked the following questions: (1) What type of prey is consumed by the hatchlings? (2) Which dietary items have high relative importance during three size intervals of hatchling Morelet’s in the hatchling–juvenile transition? (3) What is the dietary diversity and overlap in composition during the three size intervals? (4) What differences exist between the items consumed by hatchlings?

## 2. Materials and Methods

### 2.1. Study Site

The Laguna de las Ilusiones Ecological Reserve is in the center–north zone of Villahermosa, Tabasco, Mexico (18°01′ N y 92°55′ W, Figure 1). The area is 259.2 ha. The predominant climate is warm and humid with abundant rains in summer. The average annual temperature is 28.1 °C with an average annual rainfall of 2338 mm [12]. The Laguna de las Ilusiones is immersed in a growing urban area and is highly degraded. However, a permanent population of Morelet’s crocodiles inhabits this lagoon, whose density is increasing [13].

We recorded two sites (A and B) at the Laguna de las Ilusiones based on the presence of nests in previous years (Figure 1). Site selection ensured the presence of hatchlings as crocodiles are highly philopatric to their nesting sites [14,15].

### 2.2. Diet Record in Morelet’s Crocodiles

We conducted fieldwork in February, May, and July of 2018. We performed night searches on two consecutive days per month from 20:00 to 04:00 h. We carried out surveys on an aluminum boat with a 15 HP outboard motor. We used hand lamps (500 lm) and LED light headlamps (1000 lm) to locate the hatchlings. We captured hatchlings by direct manipulation, holding the hatchling by the neck with one hand and immediately holding the rest of the body to immobilize the animal [16]. We measured all captured individuals’ total length (TL) with a graduated tape measure (precision 1 mm). We considered the hatchlings’ size (TL from 300 ≤ 509 mm) following the Domínguez-Laso classification [17] to *C. moreletii*. We classified TL into three size intervals using the following formula: upper limit−lower limit/class intervals. Initial size intervals: 300–369.6 mm, *n* = 10 hatchlings; medium: 369.7–439.3 mm, *n* = 7; final 439.4–509 mm, *n* = 14). We then used the size intervals to determine the hatchlings’ relative importance index (RII), diversity, and overlap in their diet.

We carried out stomach-flushing of the Morelet’s crocodile hatchlings to obtain food samples using the Heimlich hose maneuver method [18]. We introduced the end of a nasopharyngeal probe into the crocodile’s snout, passing through the esophagus until it reached the individual’s stomach. The other end of the probe was connected to a half-gallon RL Flo-Master^®^ (Lowell, MA, USA) water sprayer. Once the probe reached the stomach, water was injected, gradually filling the stomach. We monitored its distension. Simultaneously, we applied the Heimlich maneuver by placing the crocodile head-down and forcing the extraction of the stomach contents into a 10 cm diameter strainer with a 0.5 mm sieve. We repeated the same procedure three times until there was no solid material left in the stomach. We placed the samples in hermetically sealed bottles with 70% ethyl alcohol to preserve the gastric contents and identify the items of the diet [19]. We inspected the items collected from the stomach contents under 1.6x magnification using a Carl Zeiss AG^®^ Stemi DV4 (Jena, Germany) model stereoscope. Considering the level of damage to the dietary items, we identified the items collected taxonomically to the order level for invertebrates and to the class level for vertebrates.

We calculated the numerical percentage (%N) of prey following the formula:%N = N_i_/N_t_,(1)
where N_i_ is the number of each prey item and N_t_ is the total number of prey items in the diet of each size range of hatchlings.

We estimated the volume of the components obtained from the stomach contents using the volumetric displacement method in 5 mL of water with a graduated cylinder at 0.1 mL intervals [20,21]. We calculated the volumetric percentage (%V) of each item of the three hatchlings size intervals, following the formula:%V = vd × 100/vTd,(2)
where vd is the volume displaced by each component and vTd is the sum of the volume displaced by all the components.

We used two different methods to determine the relative importance of each item in the diet: (1) The frequency of percentage incidences (%FO), which expresses the frequency of appearance of a certain component in the diet relative to the number of stomachs examined for each hatchling size intervals, according to the formula:%FO = n/N × 100(3)
where n is the number of times a component of the diet appears, and N is the total number of stomachs analyzed. (2) The relative importance index (RII) of the items constituting the diet of hatchling Morelet’s crocodiles per size intervals, following Olaya-Nieto et al. [22]:RII = %FO × %V/100,(4)

The RII intervals ranged from 0 to 100, where 0–10% represents the trophic groups of low relative importance, 10–40% represents those of secondary relative importance, and 40–100% represents the groups of high relative importance.

### 2.3. Statical Analysis

We applied a resampling approach (bootstrap with 1000 iterations) followed by a chi-square goodness-of-fit test by item to assess differences between each item consumed and the hatchling’s total length. Standardized residuals indicate the difference between observed and expected frequencies regarding standard deviations. Absolute values rather than 1.96 are considered significant at the 0.05 level. We used a logistic regression model that considered the presence and absence of items in the stomachs analyzed to determine the relationship between the items consumed individually and the TL of the hatchlings. We only considered the food items with an RII ≥ 1.0 for the analysis. We applied a Simpson diversity index to determine dietary diversity and a Pianka Index to determine the overlap in diet between hatchlings’ size intervals. We performed all analyses in the R program version 4.3.3 [23] using the packages vegan [24] and ggplot2 [25], executed through the RStudio Version 1.1.442 platform [26]. All tests were performed with a 95% confidence interval.

## 3. Results

### Diet in Hatchlings Morelet’s Crocodiles

We captured 31 hatchling Morelet’s crocodiles with an average TL of 415 mm (minimum = 300 mm, maximum = 509 mm), an interval considered Class I (590 mm) [27]. We found significant differences in the type of prey consumed by hatchlings (X^2^ = 114.12, *p* < 0.001; Figure 2). The data showed that hatchlings consumed more Coleoptera, Hemiptera, and Aranae than other dietary items. We found that less consumed items were fish, birds, Odonata, Diptera, and Scorpiones.

There were no significant differences in the probability of the consumption of Coleoptera (X^2^ = 0.001; *p* = 0.85), Hymenoptera (X^2^ = 0.006; *p* = 0.246), and Araneae (X^2^ = 0.004; *p* = 0.445) when increasing the body size of the individuals. However, although we did not find differences, Blatellids (X^2^ = 0.012; *p* = 0.125) and fish (X^2^ = 0.012; *p* = 0.148) increased in the probability of appearing as prey in the diet of hatchlings with larger body sizes, while the probability of appearance of Hemiptera decreased (X^2^ = 0.01; *p* = 0.101; Figure 3).

We washed the stomachs of all the captured hatchlings and were able to identify 11 items that made up the diet of hatchling Morelet’s crocodile, with invertebrates (Insecta, Arachnida, and Gastropoda) having the highest relative frequency (n = 9) in contrast to vertebrates (n = 2; fish, birds). The prey category with the highest displacement and volumetric percentage was Coleoptera: 13.7 mL and 87.1%, respectively. The frequency percentage of occurrence was highest for Coleoptera (87.1%), followed by Hemiptera and Araneae (74.2%); these were the items with a secondary RII, in contrast to the rest of the items that appeared with a low RII (Table 1).

The diversity of items consumed increased for the initial (D = 0.7539), medium (D = 0.8229), and final size intervals (D = 0.8715). We found significant differences between the initial and final size intervals (t = 3.3507, *p* = 0.0016) but not between initial vs. medium (T = 1.4849, *p* = 0.1436) and medium vs. final (T = 1.3233, *p* = 0.1950).

We found a high overlap in the dietary niche between individuals of the initial and medium size intervals (*O_jk_* = 0.7251) and the medium and final size intervals (*O_jk_* = 0.7279). We only found moderate overlap between the initial and final size intervals (*O_jk_* = 0.5732; Figure 4).

## 4. Discussion

The diet of hatchling Morelet’s crocodiles consisted mainly of invertebrates (insects and arachnids). This agrees with what was reported by Platt et al. [2]. López-Luna et al. [27] documented insects as the main item of the diet of Class I Morelet’s crocodile hatchlings (≤590 mm) in the same population of Laguna de las Ilusiones. However, our data include the presence of arachnids, birds, and gastropods, which they consider to be items of the diet of Class II (600–990 mm) and Class III (1000–1490 mm) crocodiles. In Belize, the diet of hatchling Morelet’s crocodile consists of six types of prey: insects, arachnids, gastropods, crustaceans, fish, and anurans, with insects and arachnids also being the main prey [2,28]. However, birds were not observed in the diet until the crocodiles reached the size of large juveniles (590–1000 mm). The diet items of hatchling Morelet’s crocodile that we reported here are similar to those reported for hatchlings of other crocodile species, such as *Crocodylus acutus* and *Caiman crocodilus chiapasius* in Mexico [10,11], *Crocodylus porosus* in Malaysia [7], and *Caiman latirostris* in Uruguay [29].

The presence of birds in the stomachs of hatchling Morelet’s crocodiles is an unprecedented event in the diet of crocodiles less than 500 mm in length. In the riparian vegetation of urban lagoons in Tabasco, there are various bird species nests [30]. Birds actively nest in areas where crocodiles are abundant to take advantage of the protection that crocodiles inadvertently provide by their presence [31]. Crocodiles intimidate and scare away small bird predators such as raccoons or opossums, and in return, the crocodiles are able to access a secure food source when chicks fall out of or are thrown from the nest, increasing the likelihood of birds being consumed by crocodiles [32]. This predatory interaction occurs with adult crocodiles that are experienced hunters on the shores of water bodies [33]. However, in the case of crocodile hatchlings, bird consumption may be more fortuitous, as when they fall from the nests near water bodies, the chicks may drown and be partially eaten by different carnivores. This behavior is analogous to that observed in some animals that do not hunt birds, such as the green iguana, which can take advantage of consuming chicken remains as carrion [34].

The variety of prey consumed by hatchling Morelet’s crocodiles in Laguna de las Ilusiones suggests generalist foraging behavior. The predominance of Coleoptera, Hemiptera, and Araneae as prey items in the diet may be because these organisms are generally found on the surface or the edge of water bodies, facilitating their consumption by hatchlings, which overlap in these foraging areas (surface of the water and vegetated strips close to water bodies) [3]. In addition, the richness and abundance of Coleoptera are higher between February and July [35], the period corresponding to our data collection months. For example, during dry and rainy periods, insects become more accessible in areas with flooded vegetation, so hatchling crocodiles forage for food and consume large amounts of prey in nearby sites, potentially reducing the energy expenditure dedicated to finding food [36].

Interestingly, we observed a decrease in the consumption of Hemiptera in final hatchling size intervals, which may correspond to the low abundance of these organisms at the end of the dry season and during the rainy season (May to October) [37]. Likewise, Hemiptera possesses various chemical substances as defensive compounds [38], which makes them palatably unpleasant to their predators. This has been documented mainly in fish [39,40] and in the lizard *Anolis carolinensis voight* [41]; however, this could be happening similarly with crocodiles since, with the passage of time and the experience acquired throughout their development, they learn to discriminate and avoid toxic or unpalatable prey through taste [42].

Our study improves our understanding of the hatchling Morelet’s crocodiles’ diet and supports previous studies showing that insects and other invertebrates comprise most of the diet. In addition, our results show differences in dietary diversity during the growth of the hatchlings, including vertebrates during the growth. Therefore, we only observed dietary overlap between hatchlings of adjacent size intervals. The ontogenetic change in the diet in hatchling Morelet’s crocodiles is like that reported in other crocodilians [43]. To our knowledge, this is the first study focusing exclusively on the dietary variation across crocodilian hatchling size intervals.

## 5. Conclusions

Our results show that the diet of hatchling crocodiles is generalist, consisting mainly of invertebrates and insects, but also includes arachnids and gastropods. We also found fish and birds as a part of dietary items not previously recorded in the hatchling crocodile diet. We, therefore, provide evidence that hatchling crocodiles less than 509 mm in length consume birds and fish, which, together with blattellids, are prey items that increase the diversity in the diet according to the size (369.6 to 439.4 mm), while some insects (such as hemipterans) decrease during the growth of the hatchling crocodile. Thus, as size intervals increase, dietary overlap changes, showing minor overlap between the initial and the final size intervals. Understanding the dietary changes during crocodile development is important in interpreting their role in food webs and ecological interactions.

## Figures and Tables

**Figure 1 animals-14-02610-f001:**
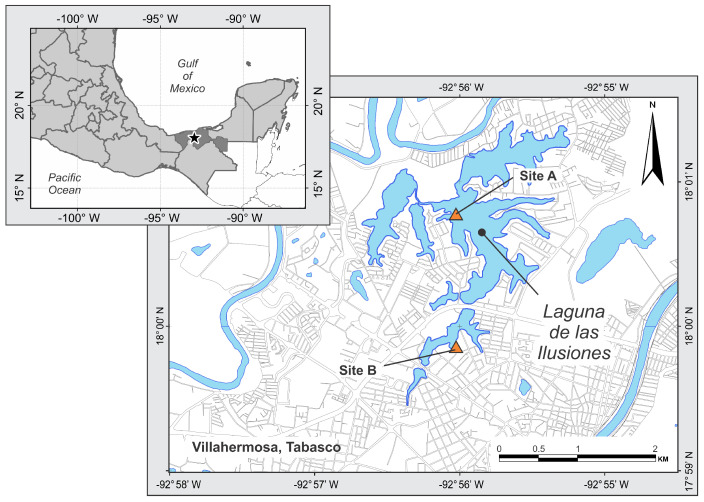
Location of the recording sites (A and B) at the Laguna de las Ilusiones.

**Figure 2 animals-14-02610-f002:**
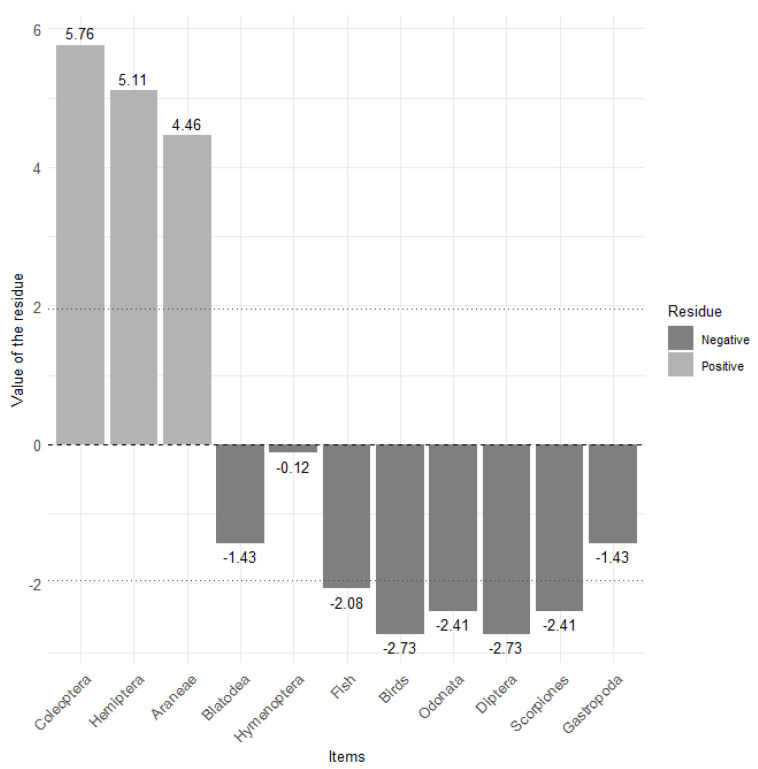
Item consumption by hatchling crocodiles. Bars extending beyond the dotted lines (−1.96 or 1.96) indicate statistically significant residuals.

**Figure 3 animals-14-02610-f003:**
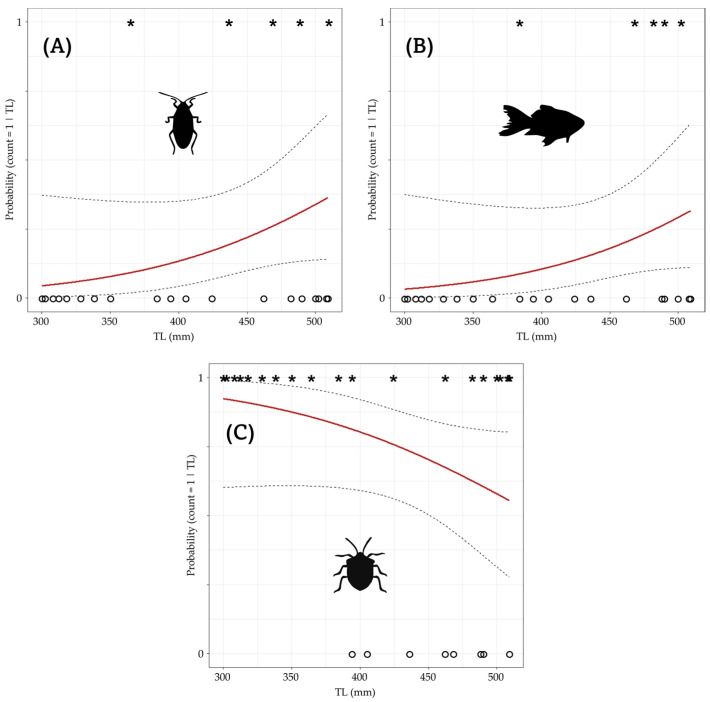
Probability of the appearance of (**A**) Blatellids, (**B**) fish, and (**C**) Hemiptera in the stomach contents concerning crocodiles’ body size (TL). The asterisk showed the presence of the item and the circles showed the absence. The red line shows the regression and the dotted lines the confidence intervals.

**Figure 4 animals-14-02610-f004:**
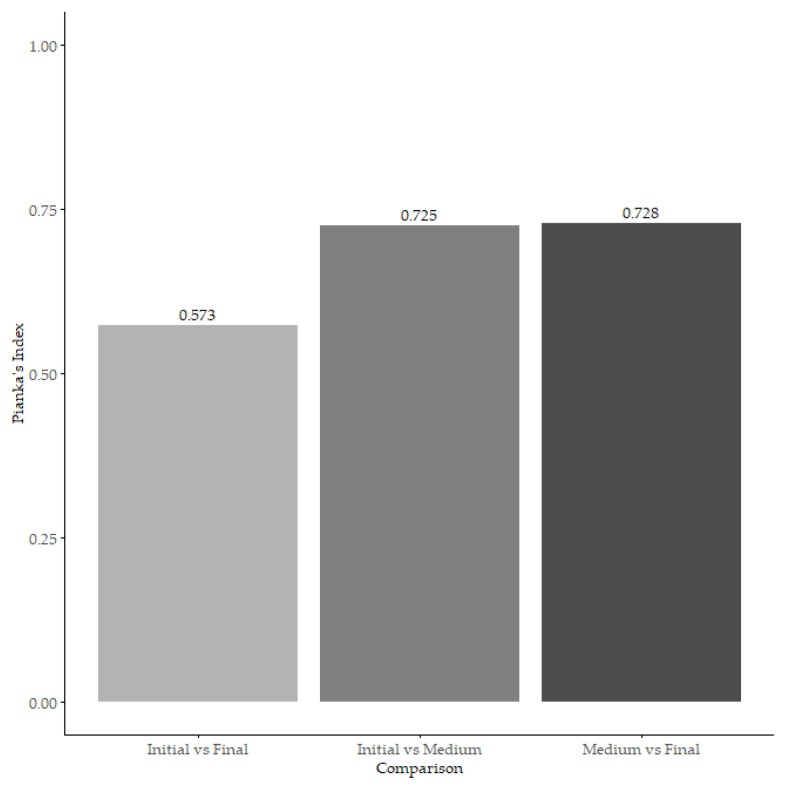
The bars indicate the overlap in the dietary items according to the size intervals of the hatchlings.

**Table 1 animals-14-02610-t001:** Numeric percentage (%N), displaced volume (VD), volumetric percentage (%V), percentage of frequency of occurrence (%FO), and relative importance index (RII, L = low, S = secondary, and H = high) of each item recovered from the stomach contents of Morelet’s crocodile hatchlings by size intervals in Laguna de las Ilusiones, Tabasco.

Size Range	Item	%N	VD	%V	%FO	RII	L	S	H
300–369.6 mm	Coleoptera	26.6	3.5	34.5	90	31.1		*	
	Hemiptera	29.4	2.1	21	100	21		*	
	Araneae	23.5	1.3	12.5	80	10		*	
	Blattodea	2.9	0.5	5	10	0.5	*		
	Hymenoptera	5.9	0.2	2	20	0.4	*		
	Fish	0	0	0	0	0	*		
	Birds	0	0	0	0	0	*		
	Diptera	2.9	0.9	9	10	0.9	*		
	Odonata	2.9	0.3	3	10	0.3	*		
	Scorpiones	5.9	0.2	1.5	20	0.3	*		
	Gastropods	0	0	0	0	0	*		
369.7–439.3 mm	Coleoptera	25	5.4	77.4	85.7	66.3			*
	Hemiptera	16.7	2.3	32.9	57.1	18.8		*	
	Araneae	25	2.1	30	85.7	25.7		*	
	Blattodea	4.2	1.3	18.6	14.3	2.7	*		
	Hymenoptera	12.5	1.3	17.9	42.9	7.7	*		
	Fish	4.2	0.3	4.3	14.3	0.6	*		
	Birds	0	0	0	0	0	*		
	Diptera	4.2	0.1	0.7	14.3	0.1	*		
	Odonata	4.2	0.5	7.1	14.3	1	*		
	Scorpiones	4.2	0.2	2.9	14.3	0.4	*		
	Gastropods	0	0	0	0	0	*		
439.4–509 mm	Coleoptera	22.2	4.9	34.6	85.7	29.7		*	
	Hemiptera	16.7	2.8	20	64.3	12.9		*	
	Araneae	16.7	3.7	26.1	64.3	16.8		*	
	Blattodea	7.4	2.1	15	28.6	4.3	*		
	Hymenoptera	11.1	1.4	10	42.9	4.3	*		
	Fish	7.4	2	13.9	28.6	4	*		
	Birds	1.9	1.3	9.3	7.1	0.7	*		
	Diptera	3.7	0.3	1.8	14.3	0.3	*		
	Odonata	1.9	0.1	0.7	7.1	0.1	*		
	Scorpiones	3.7	0.3	1.8	14.3	0.3	*		
	Gastropods	7.4	0.2	1.4	28.6	0.4	*		

* RII indicator where is the representative in the column.

## Data Availability

The data included in this study are the property of the universities UV and UJAT and can be obtained by contacting the corresponding author at jormorales@uv.mx upon request.

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
