# Peer review of "Variation in the Diet of Hatchling Morelet’s Crocodile (Crocodylus moreletii) in the Wild"

_animals, 2024, doi:10.3390/ani14172610_

Round 1

Reviewer 1 Report

Comments and Suggestions for Authors

Dear authors,
In general, the article is well structured, but there are points that must be corrected to improve the manuscript. My suggestions are as follows:

Line 02. Change “…Morelet’s crcocodile (Crocodylus moreletii) hatchlings...” by “…hatchlings Morelet’s Crcocodile (Crocodylus moreletii)…”. Review the manuscript and do the same.

Line 13. Change “growth” by “size”,

Lines 15 and 28. Keep the format. The first time said, “swamp crocodiles” and the second “Morelet's crocodile”.

Line 58. Rewrite this sentence: “following the following questions:”.

There are many spelling and grammar errors that make the methodology section very difficult to understand. Furthermore, more detailed methodological information and cited literature are needed.

Line 67. What is the climate classification system?

Line 70. Review the manuscript and use italics for scientific names.

Line 79. Change “two consecutive monthly days” by “two consecutive days at month”.

“tours by water aboard” What does it mean?

Line 80. Change “pups” by hatchlings.

Lines 88 and 89. Were animals immobilized? This is essential to prevent injuries resulting from sudden movements. Change “flexible plastic hose” by “nasopharyngeal probe”.

Line 96. How you know the stomach was empty?

Line 97. Cold alcohol?

Line 122: Due to the small size of your dataset (n = 31), a logistic regression model is not the most appropriate statistical test for contrasting the hypothesis. This is the main reason why there are no significant differences in the content of the diet components across different sizes of crocodiles. As mentioned in lines 146 and 147, you did not create intervals based on the size of the hatchlings. As you suggested in line 214, it would be better to create intervals based on hatchling size and apply the corresponding statistical tests.

Previously, in line 78, you mentioned, “We did the fieldwork during February, May, and July.” This means that you sampled in two different seasons: dry (February) and rainy (May – July). This is a methodological mistake, as it introduces a new variable into your model that interferes with your results and analysis.

Line 130. £?

Line 139. Review the title of table 1 and change the order between Diptera and Odonata.

Lines 164-167. Review this affirmation. It is not exactly that said that bibliography. I suggest change it.

According this review, I suggest conducting a new analysis of your dataset and rewriting your discussion and conclusion based on the new results.

Comments on the Quality of English Language

An extensive revision and correction of the English language are needed. There are many spelling and grammar errors that make the manuscript very difficult to understand

Author Response

Dear Reviewer,

Thank you for the opportunity to improve our manuscript entitled “Variation in the diet of hatchings Morelet´s crocodile (Crocodylus moreletii)”. The notes below refer to our revisions made considering comments added to the document. We thank for the numerous comments and suggestions, which have greatly improved our paper. 

Response to Reviewer 1

Tittle

L2. Change “…Morelet’s crcocodile (Crocodylus moreletii) hatchlings...” by “…hatchlings Morelet’s Crcocodile (Crocodylus moreletii)…”. Review the manuscript and do the same.

As the reviewer suggests, the word "hatchlings" is already in front of Morelet's crocodile  (L2).

 Simple summary

L13. Change “growth” by “size”,

 We have already changed the word growing by size (L13).

Abstract

L15-28. Keep the format. The first time said, “swamp crocodiles” and the second “Morelet's crocodile”.

 As the reviewer suggests, we have already changed Morelet´s crocodile in all manuscript.

 Introduction

L58. Rewrite this sentence: “following the following questions:”

We have removed the typing error (L55).

Materials and Methods

There are many spelling and grammar errors that make the methodology section very difficult to understand. Furthermore, more detailed methodological information and cited literature are needed.

We appreciate your comment, we have already made a revision to the document, and we consider that this latest version is according the observation you made.

 L67. What is the climate classification system?

In the document, we used a climate classification, which shows the climatological conditions reported for the study area, to help readers understand the conditions at the Laguna de las Ilusiones (L64-65).

 L70. Review the manuscript and use italics for scientific names

Thank you for the observation, we already changed it in the document.

L79. Change to consecutive monthly days by to consecutive days at month.

We already did this change at the document. We appreciate the correction (L76).

L79. Tours by water aboard what does it mean?

We sorry for the confusion, we changed to “We carried out surveys on aluminum boat” (L76).

L80. Change pups by hatchling

We already changed as you suggest (L78).

L88-89. Were animals immobilized this is essential to prevent injuries resulting from sudden movements. Change flexible plastic hose by nasopharyngeal probe.

We already described in the text that we immobilize the animal (L91-92).

L96. How you know the stomach was empty?

We add the text: “We repeated the same procedure three times until there was no solid material left in the stomach.” To validate that we did not find any solid material in the crocodile’s stomach (L97-98).

L97. Cold alcohol

We add that we used ethyl alcohol (L99).

L122. Due to the small size of your dataset (n = 31), a logistic regression model is not the most appropriate statistical test for contrasting the hypothesis. This is the main reason why there are no significant differences in the content of the diet components across different sizes of crocodiles. As mentioned in lines 146 and 147, you did not create intervals based on the size of the hatchlings. As you suggested in line 214, it would be better to create intervals based on hatchling size and apply the corresponding statistical tests.

Thank you for your comment. We decided to keep the model the same as the LGM is more accurate due to the amount of data. However, following your suggestions, we have included intervals to test for diversity and overlapping indices (L132-135).

L78. Previously, in line 78, you mentioned, “We did the fieldwork during February, May, and July.” This means that you sampled in two different seasons: dry (February) and rainy (May – July). This is a methodological mistake, as it introduces a new variable into your model that interferes with your results and analysis.

Although we understand your point, in our work, we did not compare seasons. We consider our data as a line in time.

Results

L130.  £?

Sorry for the typo. We have already corrected it (L141).

L139. Review the title of table 1 and change the order between Diptera and Odonata.

Thank you for your observation. We have already made the change (L169-174).

L164-167. Review this affirmation. It is not exactly that said that bibliography. I suggest changing it.

Thank you for your observation. We already changed, and we found a more accurate bibliography to supply (L190).

Comment. According this review, I suggest conducting a new analysis of your dataset and rewriting your discussion and conclusion based on the new results.

Dear reviewer, we understand your point, and we agree with your observation, on the table 1. We have grouped the results considering the size intervals of the hatchlings (Table 1). We also included the Pianka index, Simpson index (L175-181) and added graphs to better describe the significance of the consumed items (see Figure 2).

Comment. An extensive revision and correction of the English language are needed. There are many spelling and grammar errors that make the manuscript very difficult to understand.

We agree with your observation. We have made extensively revised the spelling and grammar of the manuscript for a better understanding.

We appreciate your time, effort and comments to improve our manuscript. In addition, we relied on the review of an English speaker to revise the manuscript.

Reviewer 2 Report

Comments and Suggestions for Authors

This is a simple work on the diet of this crocodile species that fills a gap in knowledge about its diet in Mexico. Methodology ies too much simple and sometimes not well explained. Why authors use a generalized linear model (GLM) with a binomial  distribution, using as factors the categories of registered food resources and the size range  of crocodiles. Simple range regression is sufficient. On the other hand, X2 between prey type and size classes could be estimated by means of overlap index of Shoenner or Pianka, more ussually used in diet analyses.

Line 98: This cited article describes one parasit of Crocodrylus acutus. Authors can't reference it as a method of diet identification.

I would like the authors to say up to which taxonomic level they have reached in the identification (family, genus, species?). And justify why Blattodea  is considered in the same level that fishes. Usually trophic studies homogenize the taxonomic levels or justify it-

Lines 102-116: The indices used in the diet analysis are: %V = vd *100 / vTd; %FO = n/N * 100 (Ususally named frequency or nueric percentage in diet papers) ; IIR = %FO * %V / 100% and IIR = %FO * %V / 100%.  However, other indices, frequently used in trophic analyses, are not analysed.  The %P (Percentage of presence) Number of stomachs in one prey type is present in % and sometimes even indices such as lambda combining %FO and %P (Lambda= Sum of pi2). 

Where pi is the probability that a prey belongs to a certain taxon in a stomach. 

No index of trophic niche breadth (Levins 1968) or diversity has been used either.  

Neither mean individual diversity nor cumulative diversity (Shannon or Margalef indices) has been estimated. 

Finally, it is not clear to me how volumes have been estimated. Some prey may be fragmented or semi-digested. Has this been taken into account?

The size of the prey has not been measured either, so it has not been possible to estimate the selected size and its importance in biomass by calculating the indices, not on the basis of numbers, but on the basis of the dry weight of the prey, which would have been much more indicative of the energetic importance of each of the prey. 

%P. Percent presence (named %FO in the manuscript): percentage of individuals in the population that have consumed each prey item.

%N (not included in the manuscript). Numerical percentage: the proportion of each prey item in the total number of prey items in the diet of the population. This estimator must be calculated.

%B. Biomass percentage (not included in the manuscript): the proportion of each prey item out of the total dry weight of prey in the population. Need to know dry wheight see (Wet and Dry-weight Estimates of Insects and Spiders Based on Length. https://www.jstor.org/stable/2425505 or https://doi.org/10.2307/2425505)

Lambda (not included in the manuscript): indices such as lambda combining %FO and %P (Lambda= Sum of pi2). 

For the analysis of diversity, the Shannon index applied to diet was used, f The descriptors used are:

Hi. Individual diversity; average prey diversity in each stomach.

Hz. Total Population diversity; diversity estimated by Jacknife methodology. 

See: HERPETOLOGICAL JOURNAL, Vol. 8, pp. 1 6 1 - 1 65 ( 1 998) SEXUAL AND SIZE-RELATED DIFFERENCES IN THE DIET OF THE SNAKE NATRIX MA URA FROM THE EBRO DELTA, SPAIN. XAVIER SANTOS AND GUSTAVO A . LLORENTE. 

or: DOI: 10.1002/iroh.200510899 for more complete statistical analyses.

Line 111: RII or IIR

Table 1: The number of items (n) of each prey type considered must be included in the table. Also the total number of preys in the text.

lines 121-124: I don't understand how authors analyse "the components consumed individually and the body size of the crocodiles, a logistic regression model was used" Athors need to explain more accurately. 

125: Reference of R core team.

Line 147 Where are described the size intervals considered? Authors don't indicate this.

Table 2: Main effects? What explain this table? Main effects of what analyses?

Lines 144-147: Where do the chi-square values ​​come from? How have they been estimated? They don't explain it in the methodology.

Line 152: What body size intervals are compared? it's not explained in methodology.  FIgures 1 and 2 show probabilities in continuous not by size intervals.

Figure 2 and 3: Need to explain that are red line and ppointed lines. Also authors must explain y axis.

Line 165: Add also Platt el al 2006  and Lopez-Luna et al 2019 [2, 23]

Line 176-177: This is why it is so important to have the size of prey and to be able to assess the importance of a prey in terms of biomass. There are numerous reference bibliographies where dry weight values of the detected prey can be found. On the other hand, at no point do the authors indicate the degree of degradation or digestion of the prey in the stomach, so the analysis of the volume of the diet seems to me to be not very objective. Does a digested prey count the same as a whole one? How have the authors solved this problem?

Lines 194-196: It's not clear to me that this reference says this. They can indicate the page or delete it.

Lines 200-203: 33. Bouragaoui, Z., Aba, W.B., Nouria, S. Diet of the lacertid lizard Psammodromus algirus in north Turisa Tunisia [33]. Delete this reference. Species and habitat are not comparables whith crocodyle analised habitat. 

The references are not in format and DOI is not writed in the articles referenced.

In conclusion, I consider that the paper lacks a deeper analysis of the data (many indices commonly used in diet studies are missing), the methodology is unclear, especially with regard to the statistical analyses, and I consider the sample size to be small. If we also consider that the specimens have been captured in two different areas (we do not know how many in each), perhaps a comparison between the two lagoons would have been good. On the other hand, we do not know if the two sampling points have different habitats. 

Finally, I consider that the study needs a thorough remodelling, estimating indices such as frequency (%N), lambda, niche breadth and individual and population diversity.

I therefore recommend "Major revisioN".

Author Response

Dear Reviewer,

Thank you for the opportunity to improve our manuscript entitled “Variation in the diet of hatchings Morelet´s crocodile (Crocodylus moreletii)”. The notes below refer to our revisions made considering comments added to the document. We thank for the numerous comments and suggestions, which have greatly improved our paper. 

Response to Reviewer 2

Comment. This is a simple work on the diet of this crocodile species that fills a gap in knowledge about its diet in Mexico. Methodology ies too much simple and sometimes not well explained. Why authors use a generalized linear model (GLM) with a binomial distribution, using as factors the categories of registered food resources and the size range of crocodiles. Simple range regression is sufficient. On the other hand, X2 between prey type and size classes could be estimated by means of overlap index of Shoenner or Pianka, more ussually used in diet analyses.

We appreciate your comment. We added the Pianka index, and we also used the Simpson index to know the overlap and diversity in crocodile’s diet by size intervals. This analysis can be seen on lines 175- 181 of the manuscript. We also added some graphs to better illustrate our results.

 L98. This cited article describes one parasite of Crocodylus acutus. Authors can reference it as a method of diet identification.

Thank you for your comment, we agree, and have changed the bibliography (L100).

Comment. I would like the authors to say up to which taxonomic level they have reached in the identification (family, genus, species?). And justify why Blattodea is considered in the same level that fishes. Usually, trophic studies homogenize the taxonomic levels or justify it-

Dear reviewer, we made changes to the manuscript, and we added the following text: “Consider the level of damage to the dietary items, we identified the items collected taxonomically to the order level for invertebrates and to class level for vertebrates”. We hope this clarifies your comment (L102-103).

Comment. L102-116: The indices used in the diet analysis are: %V = vd *100 / vTd; %FO = n/N * 100 (Usually named frequency or numeric percentage in diet papers) ; IIR = %FO * %V / 100% and IIR = %FO * %V / 100%.  However, other indices, frequently used in trophic analyses, are not analysed.  The %P (Percentage of presence) Number of stomachs in one prey type is present in % and sometimes even indices such as lambda combining %FO and %P (Lambda= Sum of pi2). 

Where pi is the probability that a prey belongs to a certain taxon in a stomach. 

No index of trophic niche breadth (Levins 1968) or diversity has been used either.  

Neither mean individual diversity nor cumulative diversity (Shannon or Margalef indices) has been estimated. 

Dear reviewer, we have not included the percentage of occurrence (%P) because of methodological complications (semi-digested animal parts), which do not guarantee that a single part of an animal in the stomach belongs to the same individual. However, we have included the numerical percentage (%N), the volume of the components (%V), the frequency of the percentage of occurrence (%FO) and the relative importance index (RII). We have included two indices: diversity (Simpson index) and overlap (Pianka index). Taking into account your comments, we consider that we have solved your observation.

Comment. Finally, it is not clear to me how volumes have been estimated. Some prey may be fragmented or semi-digested. Has this been taken into account?

The volumes were estimated using the volumetric displacement method (you can check lines 109-111 in the document). We hope this is sufficient to resolve your comment.

Comment. The size of the prey has not been measured either, so its has not been possible to estimate the selected size and its importance in biomass by calculating the indices, not on the basis of numbers, but on the basis of the dry weight of the prey, which would have been much more indicative of the energetic importance of each of the prey.

Dear reviewer, we understand your point, and this would have been interesting to do. However, our aim was to describe and analyse the composition, variation, diversity and overlap in the diet of hatchling Morelet's crocodiles for three size intervals during the hatchling to juvenile period. We then consider that this is outside of our objective. We will look at biomass as an excellent opportunity to do this in the future.

Comment. %P. Percent presence (named %FO in the manuscript): percentage of individuals in the population that have consumed each prey item.

%N (not included in the manuscript). Numerical percentage: the proportion of each prey item in the total number of prey items in the diet of the population. This estimator must be calculated.

We agree and include the suggested measurement for a more complete description of the food items of hatchling Morelet's crocodiles (L105-107).

Comment. %B. Biomass percentage (not included in the manuscript): the proportion of each prey item out of the total dry weight of prey in the population. Need to know dry wheight see (Wet and Dry-weight Estimates of Insects and Spiders Based on Length. https://www.jstor.org/stable/2425505 or https://doi.org/10.2307/2425505)

We appreciate your suggestion. However, as we have said, we do not consider this to be our objective. However, we will consider biomass as an excellent opportunity to do so in further investigations.

 Comment. Lambda (not included in the manuscript): indices such as lambda combining %FO and %P (Lambda= Sum of pi2)

We appreciate the observation. However, in our study we do not consider the quality of the diet, so we believe that this index is not consistent with the aim of the study.

Comment. For the analysis of diversity, the Shannon index applied to diet was used, f The descriptors used are:

Hi. Individual diversity; average prey diversity in each stomach.

Hz. Total Population diversity; diversity estimated by Jacknife methodology. 

We have already included the Simpson index in the diversity analysis because it fits our data better (L133-134).

L111.  RII or IIR

Sorry for the typo. We have already corrected it (L121).

Comment. Table 1: The number of items (n) of each prey type considered must be included in the table. Also the total number of preys in the text.

As mentioned above, it is difficult to accurately determine the number of individual prey items due to methodological complications caused by the degree of prey decomposition. Often organisms were identified by parts, making it uncertain whether they belonged to the same individual. We have therefore chosen to indicate only the presence or absence of each item.

L121-124. I don't understand how authors analyse "the components consumed individually and the body size of the crocodiles, a logistic regression model was used" Athors need to explain more accurately. 

We apologise for the confusion. We have tried to improve the wording of the idea for better understanding in the text (L26-137).

L125. Reference of R core team.

We have already made the appropriate citations (L135-137).

L147. Where are described the size intervals considered? Authors don't indicate this.

Thank you for your comment. We have already included the description of how the size intervals were selected (L84-88).

Comment. Table 2: Main effects? What explain this table? Main effects of what analyses?

L144-147: Where do the chi-square values ​​come from? How have they been estimated? They don't explain it in the methodology.

Thanks to the suggestions made regarding data analysis, we decided to modify some of them. As a result, this table was removed because it did not fit the results we wanted to present and was replaced by a Chi-square goodness of fit test per item to assess the differences between crocodile sizes (see Figure 2).

L152. What body size intervals are compared? it's not explained in methodology.  FIgures 1 and 2 show probabilities in continuous not by size intervals.

Figure 2 and 3: Need to explain that are red line and ppointed lines. Also authors must explain y axis.

Figures 2 and 3 (now Figure 3) correspond to the logistic regression model in which we analysed the items consumed individually and the total length of the crocodiles. The size intervals were used to determine the relative importance index (RII) of the diet, diversity and overlap only. On the other hand, we have already included the explanation and modified the red line and the pointed lines (L159-160).

Line 165: Add also Platt el al 2006  and Lopez-Luna et al 2019 [2, 23]

We have chosen a more precise quote that refers directly to the sentence (L190).

L176-177: This is why it is so important to have the size of prey and to be able to assess the importance of a prey in terms of biomass. There are numerous reference bibliographies where dry weight values of the detected prey can be found. On the other hand, at no point do the authors indicate the degree of degradation or digestion of the prey in the stomach, so the analysis of the volume of the diet seems to me to be not very objective. Does a digested prey count the same as a whole one? How have the authors solved this problem?

Dear reviewer, we understand your point of view. However, as we have already mentioned, we consider this to be outside our objective. On the other hand, the food volume analysis was estimated using the volumetric displacement method. Because the prey in the crocodiles' stomach contents could be in different stages of degradation, we chose this method, which measures the volume displaced by the specific component, it doesn't matter whether it's a whole prey or a part of it.

L194-196. It's not clear to me that this reference says this. They can indicate the page or delete it.

We have changed the quote to one that better reflects the idea (L221-222)

L200-203. 33. Bouragaoui, Z., Aba, W.B., Nouria, S. Diet of the lacertid lizard Psammodromus algirus in north Turisa Tunisia [33]. Delete this reference. Species and habitat are not comparables whith crocodyle analised habitat. 

We change the citation to one that is comparable to the habitat analysed (L226; L361).

Comment. The references are not in format and DOI is not writed in the articles referenced.

We appreciate the comment. We have included the available DOIs and ensured that we have followed the reference format provided in the manuscript template.

Comment. In conclusion, I consider that the paper lacks a deeper analysis of the data (many indices commonly used in diet studies are missing), the methodology is unclear, especially with regard to the statistical analyses, and I consider the sample size to be small. If we also consider that the specimens have been captured in two different areas (we do not know how many in each), perhaps a comparison between the two lagoons would have been good. On the other hand, we do not know if the two sampling points have different habitats. 

 Finally, I consider that the study needs a thorough remodelling, estimating indices such as frequency (%N), lambda, niche breadth and individual and population diversity.

Dear reviewer, we appreciate your precise observations, so we have included the suggested diversity and overlap indices. We have also tried to clarify the methodology and modified some of the statistical analyses for greater clarity. Regarding the collection of specimens, we apologise for the confusion; they were not collected from different lagoons. We recorded two sites (A and B) at Laguna de las Ilusiones, which we have already specified in the text (L69).

We appreciate your time, effort and comments to improve our manuscript.

Reviewer 3 Report

Comments and Suggestions for Authors

The issue of variability of the crocodile feeding pattern undertaken by the researchers is extremely interesting and complex. As the authors noted, it has a broader ecological aspect and shows the dependencies in the ecosystem occupied by crocodiles.

A few observations are made by the reviewer:

1. The introduction clearly illustrates the hypothesis tested by the researchers, has a clearly indicated aim [52-62], research assumptions and clearly poses scientific questions.

2. Chapter: material and methods described correctly and in detail with a clear indication of the method of conducting statistical analysis.

3. The results obtained are described with the correct use of links to tables and charts, but the description of the results seems to me to be insufficient. The authors could be tempted to describe it more extensively, especially since the results are encouraging.

4.The percentage description of the results [128-138] would look much better illustrated by a chart, I encourage you to edit Table 1 and enrich the manuscript with a chart.

5.The discussion is substantively conducted with correct references to the presented literature.

6.  The results are interesting, written on the basis of the analysis, but they could better correspond with the purpose and assumptions of the manuscript included in the introduction chapter (answers to research questions) 

To sum up, the manuscript is very interesting, the results of the experiment are professionally collected, analyzed and described. The manuscript recommends for printing

Author Response

Dear Reviewer,

Thank you for the opportunity to improve our manuscript entitled “Variation in the diet of hatchings Morelet´s crocodile (Crocodylus moreletii)”. The notes below refer to our revisions made considering comments added to the document. We thank for the numerous comments and suggestions, which have greatly improved our paper. 

Response to Reviewer 3

  1. The introduction clearly illustrates the hypothesis tested by the researchers, has a clearly indicated aim [52-62], research assumptions and clearly poses scientific questions.

We appreciate your comment. We have tried to convey the study's objective as clearly as possible, and as a result, we included an additional question to enrich the manuscript (L55-59).

  1. Chapter: material and methods described correctly and in detail with a clear indication of the method of conducting statistical analysis.

Additionally, we improved part of the methodological description and enhanced the statistical analyses like Pianka index and Simpson index.

  1. The results obtained are described with the correct use of links to tables and charts, but the description of the results seems to me to be insufficient. The authors could be tempted to describe it more extensively, especially since the results are encouraging.

Dear reviewer, we agree. That’s why we added the previously mentioned indices, modified Table 1, and included graphs (see figures 2-4) to better describe the significance of the consumed items and obtain more comprehensive results. Additionally, we modified the description of the results based on the newly added analyses (L175-181).

4.The percentage description of the results [128-138] would look much better illustrated by a chart, I encourage you to edit Table 1 and enrich the manuscript with a chart.

We appreciate the suggestion. We already done the editing of Table 1 and added a charts to enrich the manuscript (Figures 2 and 4).

5.The discussion is substantively conducted with correct references to the presented literature.

Thanks for the comment. We added the discussion corresponding to the results of the added analyses (L245-254).

6.The results are interesting, written on the basis of the analysis, but they could better correspond with the purpose and assumptions of the manuscript included in the introduction chapter (answers to research questions) 

We agree with the suggestion. We restructured the results based on the order of the research questions posed, aiming to ensure the manuscript follows a logical sequence and thereby facilitate the reading and understanding of the study.

To sum up, the manuscript is very interesting, the results of the experiment are professionally collected, analyzed and described. The manuscript recommends for printing

We appreciate your time, effort and comments to improve our manuscript.

Round 2

Reviewer 1 Report

Comments and Suggestions for Authors

After a carefully review of the manuscript, it is notorious the improve the current version of the manuscript being suitable for its publication. There are a minor detail to correct. I would like to congratulate the authors for your effort.

Line 80-82. The sentence is duplicated. L

ine 148. Bold “Figure 2”.

Line 186.Bold “Figure 4” and include a heading for the table.

Unified terminology: Let’s consistently use ‘fish’ instead of ‘piscis’ and ‘birds’ instead of ‘aves’ throughout the manuscript.”

Author Response

Second round reviewer 1.

After a carefully review of the manuscript, it is notorious the improve the current version of the manuscript being suitable for its publication. There are a minor detail to correct. I would like to congratulate the authors for your effort.

Line 80-82. The sentence is duplicated. We already did the correction.

Line 148. Bold “Figure 2”. We already did the change

Line 186.Bold “Figure 4” and include a heading for the table. We already include the heading.

Unified terminology: Throughout the manuscript, let’s consistently use ‘fish’ instead of ‘piscis’ and ‘birds’ instead of ‘aves’.” We revised and homogenized the names.

Reviewer 2 Report

Comments and Suggestions for Authors

I think the authors have made a great effort in revising their article and that it can be accepted in its current form. However, I consider it a pity that the authors could not go deeper into the energy value of the diet from the sizes and biomass of the same.

However, my final conclusion is that it can be accepted in its current form.

Author Response

Second round reviewer 2

 I think the authors have made a great effort in revising their article and that it can be accepted in its current form. However, I consider it a pity that the authors could not go deeper into the energy value of the diet from the sizes and biomass of the same.

We appreciate the comments; we will consider the energy value, sizes, and biomass for another publication.